# The Efficiency of Drones Usage for Safety and Rescue Operations in an Open Area: A Case from Poland

**Norbert Tuśnio** [1,*] and **Wojciech Wróblewski** [2]

1   Faculty of Safety Engineering and Civil Protection, The Main School of Fire Service, 01-629 Warsaw, Poland
2   Internal Security Institute, The Main School of Fire Service, 01-629 Warsaw, Poland;
    wwroblewski@sgsp.edu.pl
*   Correspondence: ntusnio@sgsp.edu.pl

**Abstract:** The use of unmanned aerial systems (UAS) is becoming increasingly frequent during search and rescue (SAR) operations conducted to find missing persons. These systems have proven to be particularly useful for operations executed in the wilderness, i.e., in open and mountainous areas. The successful implementation of those systems is possible thanks to the potential offered by unmanned aerial vehicles (UAVs), which help achieve a considerable reduction in operational times and consequently allow a much quicker finding of lost persons. This is crucial to enhance their chances of survival in extreme conditions (withholding hydration, food and medicine, and hypothermia). The paper presents the results of a preliminary assessment of a search and rescue method conducted in an unknown terrain, where groups were coordinated with the use of UAVs and a ground control station (GCS) workstation. The conducted analysis was focused on assessing conditions that would help minimise the time of arrival of the rescue team to the target, which in real conditions could be a missing person identified on aerial images. The results of executed field tests have proven that the time necessary to reach injured persons can be substantially shortened if imaging recorded by UAV is deployed, as it considerably enhances the chance of survival in an emergency situation. The GCS workstation is also one of the crucial components in the search system, which assures image transmission from the UAV to participants of the search operation and radio signal amplification in a difficult terrain. The effectiveness of the search system was tested by comparing the arrival times of teams equipped with GPS and a compass and those not equipped with such equipment. The article also outlined the possibilities of extending the functionality of the search system with the SARUAV module, which was used to find a missing person in Poland.

**Keywords:** unmanned aerial systems; search and rescue operations; missing people; data transmission devices; automatic flight

## 1. Introduction

Every year ca. 20,000–30,000 people go missing in Poland. Most of them are found on the day of their disappearance, but approximately 4000 continue to be missing over a longer period [1]. As it comes to search and rescue operations, the key factor in this type of situation is the time required to reach the missing person. Until recently, specialist search and rescue groups used specially trained dogs or modern basic technological solutions, such as thermal imaging. Nowadays, units delegated to such actions tend to deploy a wide assortment of technological solutions, which was presented in [2] for three different operational paradigms supporting this type of action in the field, and factors that affect the determination of the best paradigm in the human–robot system were specified, which include the following.

- Sequential operations—a strategy appropriate for search operations that need to be executed in a difficult terrain, with restricted mobility, in situations with limited

available human resources operating in the field and where the likelihood of the missing person's presence is evenly distributed over a large area;

- Remote-controlled operations—suitable for situations where the mission manager has more access to relevant information from the field than from the base station;
- Base-directed operations—appropriate for areas that offer high mobility for the field search team, but not enough information that would be required to conduct a hasty search.

The location of a missing subject is crucial to allow selecting the most advantageous paradigm under the given circumstances and modelling the behaviour of missing persons in a given situation is just as important. The technologies used so far greatly contribute to reducing the time required to reach missing subjects, which is particularly significant in the case of injuries caused by falls, hypothermia, dehydration and chronic illnesses. However, those technologies need to be continuously improved to reduce the mission times to a minimum.

In Poland search and rescue operations involve a lot of specialised units, among others also the State Fire Service, which carries such operations in the basic, specialist and specialised fields of international humanitarian assistance [3]. In Poland, in situations of disappearances taking place in an open ground, the leading entity is the police, which, on the basis of a concluded agreement, cooperate with search and rescue groups both in the structures of the state and volunteer fire brigades and other civil associations, e.g., Mountain Voluntary Rescue Service [4].

Unmanned aerial systems supported by devices that provide communication and data transmission from UAVs are increasingly frequently used in search and rescue operations for missing persons. This technology is still being validated by fire protection units, among others. Consequently, an attempt was made to develop a solution to support this type of action in the open ground, which involved the Main School of Fire Service (MSFS) and Volunteer Fire Department (VFD) Niegoszowice. The undertaken analysis demonstrates the effectiveness of actions taken by rescuers using UAS and supported by a GCS workstation, which assures a local Wi-Fi network and web server, internet access, video transmission from UAVs to mobile devices and a powerful radio station. The study takes into consideration not only UAVs as such, but also the environment related to their operation and coordination of ground groups and modern devices for image transmission and communication assurance. Until now, the use of UAVs has consisted of terrain observation and image analysis, including thermal imaging. A part of the performed tasks included producing an orthophotomap composed of images taken from one or many UAVs. An analysis of the orthophotomap made it possible to determine the location coordinates of the missing subject. The effectiveness of the use of UAVs in search and rescue operations was proven (Tychy [5], Lincolnshire and Fort Wayne [6]). However, this is not the full extent of their capabilities, and the advancements in this technology confirm the increasingly frequently implemented direction, and namely the use of automated systems based on artificial intelligence [7]. One such solution is the SARUAV software, which facilitates and speeds up search and rescue operations in which UAVs are used [8]. The system is a very useful tool that can be crucial in many search and rescue operations. Mountain Volunteer Rescue Service successfully used the system during a search operation for a missing person. The incident took place on 29 June 2021 [9].

The literature indicates that similar solutions exist in the area under study, such as Loc8 software or the MOBNET system. The differences between these and SARUAV are shown in Table 1.

**Table 1.** Comparison of the capabilities of the SARUAV system and other solutions with similar functionality.

| Item | System (Software or Specialist UAV) | Description of System | Source |
|---|---|---|---|
| 1. | Search and Rescue with Unmanned Aerial Vehicle | Algorithm for searching for missing persons in undeveloped areas. The system has 2 modules for the following:<br>1. Determining the maximum human walking range in a given time.<br>2. Automatic image analysis and identification of potential human locations. | [8] |
| 2. | Loc8: Image Scanning Software for Search and Rescue | Software used to scan images or videos after entering colour that is being searched for (e.g., clothing). Finding a match is signalled by an alarm and indication of the location. The target is then verified and rescuers are dispatched to the location. It can even analyse satellite images. | [10] |
| 3. | Mobile network for people's location in natural and man-made disasters | A system for locating victims during natural disasters and crisis situations, such as earthquakes, hurricanes or major blizzards. The basic assumption of its operation is related to the fact that the searched person has a working and switched-on mobile phone. | [11] |
| 4. | Multi-task UAV | A rotary wing flying platform designed for flight in mountainous terrain at negative temperatures, high altitudes and strong winds. Equipped with an avalanche detector, cameras (daylight and thermal imaging) and various payloads (rescue kits, special explosive cartridge for controlled triggering of an avalanche). Capable of fully autonomous flight and terrain search. | [12] |
| 5. | MAGI: Multistream aerial segmentation of ground images | A fast image recognition algorithm with which, thanks to the hardware used, real-time performance can be achieved. The model is suitable for operations where time is critical, such as fire detection and search and rescue operations. | [13] |
| 6. | RGDiNet: efficient onboard object detection | A multimodal platform for real-time object detection that can be mounted on a UAV and which is insensitive to changes in the brightness of the surrounding environment. | [14] |

The analysis shows that there are several image recognition systems on the market, but only SARUAV works offer a ready-made map, which makes it possible to determine the area that could be reached by the missing subject, based on their speed and taking into account terrain difficulties.

There is also a trend to build dedicated UAS appropriately to the type of threat, but the optimal solution in search and rescue operations seems to be to use any UAV to take pictures and then send the rescue teams to the place pre-designated by the software. This is how both the SARUAV and Loc8 software are designed.

Consequently, the aim of this publication is to analyse the process of cooperation between particular services and the effects of search and rescue activities, as well as to estimate the conditions for minimising the time of arrival of rescuers to the injured person in open areas. An outline is made of UAVs and IT technologies (such as GCS workstation), which are capable of supporting the search for missing persons and should be in use by services responsible for safety and civil protection.

In the experiment carried out in Nowy Dwór Mazowiecki, during the testing phase, selected systems supporting search and rescue operations (UAVs, and command suitcase) were used. For formal reasons, the SARUAV system was not used for testing, and a concept was presented that extended the functionality of the coordinated search and rescue group with the SARUAV system. We found some license restrictions in one province due to the lack of availability of appropriate digital maps. For each launch of the system, it is necessary to prepare dedicated spatial data, and the automatic detection of people

works in a specific area "purchased" by the recipient of the system. Evidence of the effectiveness of the presented concept with the SARUAV model was demonstrated in the form of positive results of finding a living person and may constitute an effective extension of the functionality of the system used in search and rescue operations [15]. The solution uses a nested k-means algorithm to detect people in aerial photos from close proximity [16]. The software uses a variety of non-statistical detection methods [15]. The SARUAV system was trained on pictures of people wearing clothes of different colours and patterns (including smaller objects: children and dogs on the pictures). The training sets were made with the engagement of the study participants. The SARUAV system effectively detects the figures of any dressed adults and children, and the presence of dogs (it does not detect big animals). The key parameter for the SARUAV system is the ground sample distance (GSD), which depends on the flight altitude and the characteristics of the camera. It is best to fly so that the GSD is less than 3 cm / px. The two detection algorithms work in parallel, only on nadir and RGB images. The conducted near infrared (NIR) tests gave less satisfactory detection results. From the perspective of the algorithms used, the far-infrared imaging has insufficient resolution.

## 2. Methods

The conducted research focused on the characteristics, specifics and prospects of deploying UAVs in rescue operations. Tests related to the usage of UAVs in rescue operations, prepared by the VFD, took place on 8 July 2021 at the Base of Training and Rescue Innovation at the Main School of Fire Service in Nowy Dwór Mazowiecki (Figure 1).

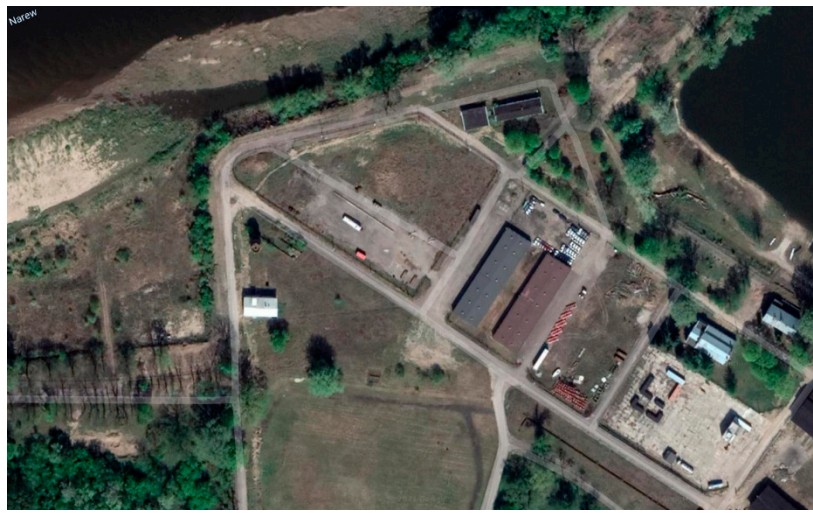

**Figure 1.** Practice area on the training grounds in Nowy Dwór Mazowiecki. Source: Google Maps.

Hardware and software components of the VFD concept system (listed in the Appendix A) were subjected to testing, as well as the possibility of using UAVs and the GCS workstation (Figure 2).

The MSFS took part in simulated operations. The participants of those simulations coordinated actions, using preview and the public address system integrated with UAVs. The tests verified the complementarity of the information exchanged, which can be used in operations, along with the level of interoperability between the SFS and the VFD.

The participants were divided into two groups with Alpha and Delta identifiers. Starting from different locations on the training ground, they were to reach a specific point according to commands received via radio from the head of the rescue operation (HRO), who directed them by comparing an orthophotomap of the terrain with their position indicated by the drones. Additionally, communication failures were simulated in the Alpha group. The HRO could only transmit commands and messages via the speaker mounted on the DJI Mavic 2 Enterprise Dual drone. However, there was no return radio channel,

and the accuracy of the executed commands was judged by the HRO on the basis of the drones' image. Two parts of this phase of the exercise were executed. In the first one, the participants of the study went into action without prior preparation and without having at their disposal any locating equipment (GPS, compass). It was not easy to coordinate their work, and the commands, directions, reference points proved to be ambiguous. Despite these difficulties, both teams were successfully led to the set-out point. At the second attempt, the participants of the exercise already knew the terrain and were equipped with locating devices; therefore, the time of reaching the target was significantly shortened.

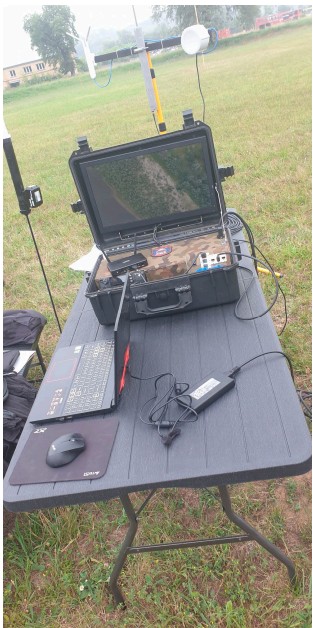

**Figure 2.** GCS workstation designed and built by VFD members. Photo: N. Tuśnio.

The experimental methods included the study of the following:

- Resilience of the radio communication system to interferences and harsh terrain conditions;
- Resistance to communication interference between UAV and the control device;
- Elements that can excite frequencies that could affect the work of UAV during operations;
- Elements that reduce the risk of losing control to an acceptable level of risk;
- The quality of images from UAV cameras and their usefulness in operations;
- The audibility and intelligibility of messages emitted from the UAV integrated public address system;
- The time required to search for a missing person without and with the system.

Tests using UAVs: AUTEL EVO II and DJI Mavic 2 Enterprise Dual were executed according to the guidelines provided in [17–19] and in internal VFD working arrangements. A simulation was carried out of search and rescue operations, during which teams were coordinated with the use of UAVs. A prototype GCS workstation was used, which allowed real-time viewing of UAVs on the HRO workstation [20].

In the first stage of the exercises, a precise orthophotomap of the area was produced. For this purpose, a UAV AUTEL EVO II with a camera of 8K resolution was used, which performed a flight at an altitude of 90 m in the mode of serial nadir images (camera directed perpendicularly to the surface of the earth) [21]. The flight route was prepared in the ladder method. This method allows a regular coverage of the area with shots overlapping each other in 50%–70% to enable the establishment of a very accurate orthophotomap of a given site, which can then serve as a basis for further operational activities. A similar method was successfully used during the Biebrza National Park fire in Poland in 2020 [22].

Three exercise scenarios were carried out as part of the research: 1—creation of orthophotomap in field conditions; 2—coordination of ground groups; and 3—terrain

observation with using a thermal imaging camera. In each case, three people controlled the flight operation (UAV operator, observer technician, and team leader).

A time measurement technique allowed recording the experiment on a video camera and then reading out the individual task times.

On the day of the experiment, the weather conditions in Nowy Dwór Mazowiecki were as follows: cloudless, light rainfall (1 mm), daytime temperature 29 °C, pressure 1019.5 hPa, and wind speed 20.25 km/h, direction south-east.

A search algorithm involving HRO targeting teams equipped with GPS and a compass was used, and tests were executed of the effectiveness of operations without this equipment and in the event of communication failure.

### 2.1. Structure and Organisation of the UAVs Section

The organisational structure of the UAVs section during operational activities arises not only from aviation law regulations (the specific role of the UAV operator), but also from actual operational needs.

The UAV operator is the person responsible for the execution of each flight and makes the final decision of whether a flight can be performed at a given location. It is his/her responsibility to carry out the technical inspection of the UAV before take-off, control it and assure safety of the mission performed.

The responsibility of a technician, also acting as an observer, is the preparation of other necessary equipment including running the GCS workstation, establishing communications with other units and observing the UAV flight. During the exercise, he/she remains with one of the cadet teams to assess their performance. The role of the technician is also to secure the area of flight operations to make sure that especially take-offs and landings are done in a manner safe for bystanders.

The team leader acts primarily as a liaison between the UAVs unit and the HRO (or other services). His/her task is to be a kind of information filter so that the operators can fully concentrate on their tasks. He/she is also responsible for coordinating the activities of various operators who, controlling UAVs in the same space, must correlate their actions so as to ensure an adequate level of flight safety.

An important issue when conducting operations involving the use of unmanned aerial systems is the time of data transmission. A method is adopted in the organisation of operational activities that introduces parallel actions upon arrival in the area of operation in the following operational algorithm:

1. Operators prepare the UAVs for launch procedure in the shortest operational time possible while maintaining relevant safety rules.
2. The technician prepares GCS workstation and activates communication systems including the LTE/5G Wi-Fi mast.
3. The flight team leader establishes a detailed action plan with the other participants involved in the operation and supervises the completeness of the launch procedures.

All this allows launching of the UAV within 3 min of arrival in the operational area. The first image in the GCS workstation can be obtained as early as in the 6th minute, and the image in the mobile devices of the participants of the operation in the 8th minute after the start of the operation.

### 2.2. Equipment Resources

The VFD unit is equipped with the following technology and software (Table 1 in the Annex A lists the equipment used during the exercises):

1. Two UAVs: DJI Mavic 2 Enterprise Dual and AUTEL EVO II.
2. PIX4Dreact—the application enables the mapping of the area of action (making an orthophotomap). On the basis of the UAV flight, an accurate and up-to-date situational map is established in field conditions.

3.  Live preview RTMP—the image from the drones is visible on monitors (laptop, GCS workstation) and on smartphones of the participants. It is also possible to transmit the image via the Internet.
4.  Internet access—a Wi-Fi MESH network with Internet access is set up. This enables information to be passed on quickly to rescuers.
5.  LOCAL WWW—the GCS workstation has its own web server, which makes it possible to record and present information that is relevant for the operation (e.g., search report).
6.  SARUAV—supports the search for missing persons through numerical modelling of movements and analysis of images from UAVs.
7.  Thermal imaging camera—one of the UAVs equipped with it, together with a telemetry system, allows precise identification of temperatures in the field of view of the camera.
8.  NFRS radio communications—high-power mobile radio with high-performance antenna that allows maintaining communications in difficult conditions.
9.  Internal radio communication—communication within the group of UAV operators takes place on dedicated equipment and frequencies so as not to cause interference or interference with other services.

Worthy of particular attention is the GCS workstation developed by VFD members (see Figure 3 for a working diagram), which allows the following:

1.  Multiplication (replication) of drone images and their transmission to other devices both on the Internet and on a local network.
2.  Creating a local Wi-Fi network to provide access to the Internet and to local information as well as video from UAVs.
3.  Communication set-up also in difficult terrain conditions, where ordinary mobile or portable radios are unable to cope with communication requirements (it is equipped with a mobile radio in the SFS radio communication standard with an antenna of high energy gain).

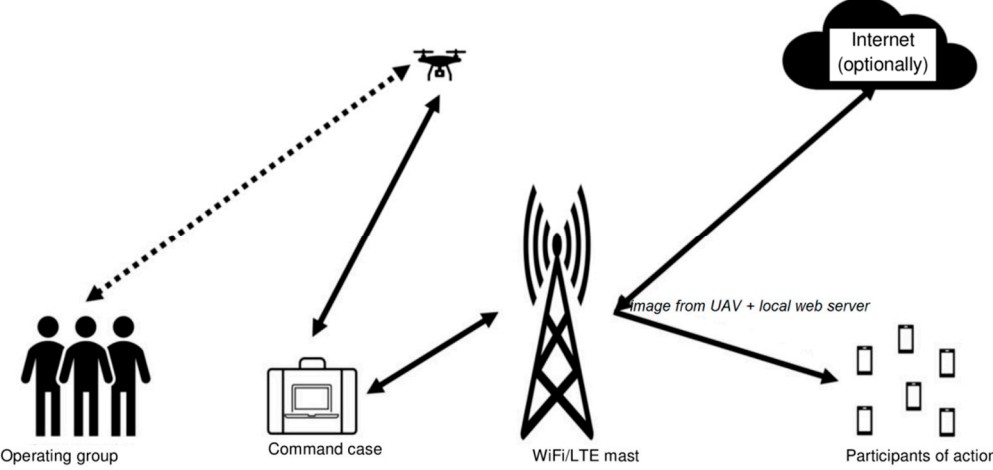

**Figure 3.** Operational diagram of an operational group equipped with UAV and GCS workstation.

A key issue in the assembly of the GCS workstation is the proper separation of all connections and the elimination of any potential radio interference.

Both the Wi-Fi network and UAVs communication operate in the 2.4 and 5 GHz standards. It is therefore important to choose the optimal slot (channel) for connections to avoid possible interference with the connection between UAVs and control equipment. This may be done by using high quality Ubiquiti network devices, which constantly verify the network occupancy.

All cables and connections used in the GCS workstation need to be shielded to minimise potential interference. A consistent power source and high quality converters and voltage stabilisers guarantee the stable operation of the entire solution.

*2.3. Selected Scenarios*

Scenario 1. Development of an orthophotomap in field conditions.

- The concentration point acts as a field command centre.
- Site mapping action is carried out to plan and support subsequent activities.
- The action area is not known to the rescuers beforehand. It was necessary to produce an orthophotomap of the area in field conditions quickly and precisely, which was essential for further actions.
- Performing a ladder flight, creating a series of photos for processing in PIX4Dreact application—UAV AUTEL EVO II was used.
- Video transmission from the UAV to the concentration point and live video retransmission to the participants' mobile devices—use of GCS workstation system.
- Estimation of distances between sites and planning access from three sides by different ground teams—using the orthophotomap created beforehand.
- Activities of two-person ground teams when choosing different routes to reach—based on the developed orthophotomap.

Scenario 2. Coordination of ground teams.

- The concentration point acts as a field command centre.
- Coordination of ground teams—follow-up action after Scenario 1.
- Ongoing monitoring of the passage of two ground teams; the Commander may relay commands via radio from the command centre to the ground teams, e.g., to modify the route of the UAVs.
- Simulation of radio communication failure in one of the groups (radio failure), transmission of the command from the Commander to the ground group using the integrated speaker system UAV DJI Mavic 2 Enterprise Dual (Figure 4).

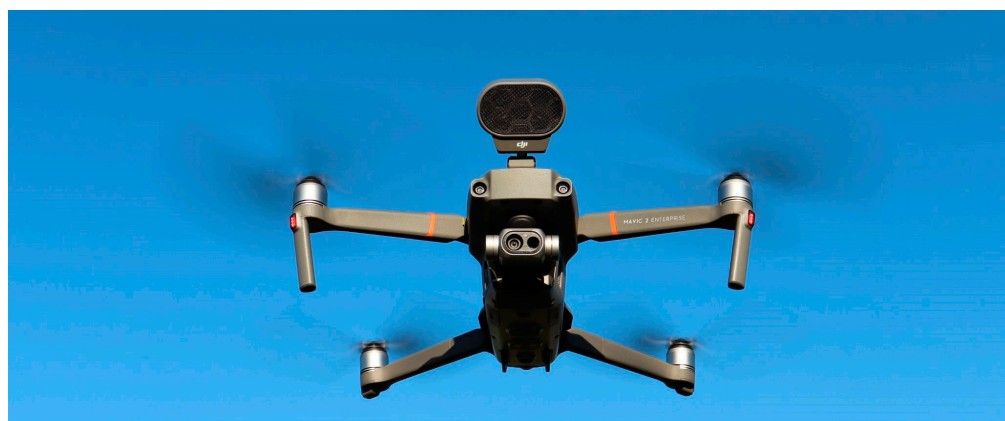

**Figure 4.** DJI Mavic 2 Enterprise Dual—UAV with mounted speaker. Photo: K. Orzepowski.

- Arrival of the first team to the operation site (selected object on the training grounds).

Scenario 3. Ground observation using a thermal imaging camera.

- The concentration point serves as a field command centre.
- A fly-around of the operations site, carrying out observations with the use of a thermal imaging camera, live viewing visible in the field command centre—use was made of a DJI Mavic 2 UAV.
- Indication by the commander by radio of particularly dangerous places to conduct operations.
- Assuming static positions by other available UAVs in designated places over the operation site, live transmission of images of implemented actions taken from different perspectives, continuous monitoring by each UAV of the assigned area (person or object).

### 3. Results

The solution adopted in Scenario 1 allowed the development of a precise, up-to-date map of operations on the entire area of over 25 ha in less than 20 min (this time should be shorter than the flight duration of the available drone). This turned out to be greatly helpful in conducting real rescue actions in an unknown area. It made it possible to determine the required forces, the route of access, key locations, etc., in a remote and therefore safe way. This method also allows assessing the differences in the image of a given area, e.g., following fires, floods, hurricanes or other disasters, which is also extremely important during search operations after building disasters caused by earthquakes.

Two battle groups, designated Alpha and Delta, were deployed to perform the task in Scenario 2. Rescuers from Section A and Section B were deployed in two locations characterised by a differentiated terrain. Both the A section and B section were tasked to reach the simulated incident site. The HRO directed the rescue teams using radio communications based on a photomap of the terrain and imagery transmitted from UAVs. A simulation was made of radio communication failure, and the integrated sound system in the DJI Mavic 2 Enterprise Dual UAV was used as a surrogate environment. However, there was no return radio channel, and the accuracy of the executed commands was assessed by the HRO based on the UAVs image. Therefore, this type of communication was unidirectional and only suitable for use in emergency situations.

Two variants of this particular phase of the exercise were carried out. In the first variant, the rescuers started executing the assigned actions without previous preparation and without locating equipment (GPS, compass). The coordination of their work turned out to be difficult (as only the direction of the needed turn had been specified) and the commands, directions and reference points proved to be quite ambiguous. Despite these difficulties, teams A and B were successfully led to the planned point.

In the second variant, the HRO and rescue teams were equipped with GPS and a compass. This allowed reaching the destination in a much shorter time, and both command and coordination proved to be more efficient.

The implementation of Scenario 3 that involved using a thermal imaging camera made it possible to further improve the effectiveness of search and rescue operations for missing persons carried out in the wilderness. In such situations, however, there are some limitations due to the fact that the camera is unable to scan solid materials. It is therefore only possible to look for the outlines of objects that actually emit heat. The thermogram only shows small, warmer spots that may indicate a person standing behind bushes and trees (Figure 5).

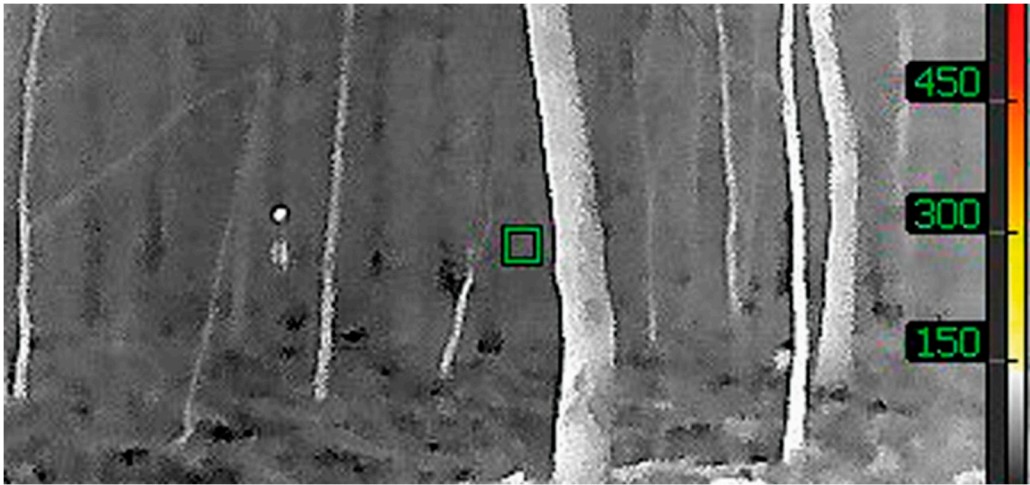

**Figure 5.** Small warmer spots visible in the thermogram (to the left of the green square), which may indicate a person standing behind bushes and a tree. Photo: W. Pruss.

The above scenarios show that the time necessary to reach a target (e.g., a lost subject) depends to a large extent on the supporting technologies used. This means that the applied solutions should be extended with new functionalities. Support in this regard can be provided by the SARUAV software, which provides a real possibility of the optimal use of drones in the search for missing people. The system enables this technology to be used quickly, efficiently and inexpensively, without the need of involving an additional group of people or other resources in the search [23].

The most important functionality of the SARUAV system is the automatic detection of people.

For example, it is possible to survey an area of ca. 3 ha with great accuracy, using a drone in approximately 10 min by taking about 150 aerial photographs with very high lateral and longitudinal coverage and high field resolution of the images (less than 2 cm/px) [23].

This set of images is automatically processed by the SARUAV system in 2–3 min, and false positives can be filtered out in further 3–5 min in a verification panel designed for this purpose.

By using two detectors with different methodological underpinnings, the system maximises the probability of human detection in an unattended mode and minimises the number of false negative readings [23].

The time needed for an automatic analysis of aerial photographs and evaluation of its results was under 10 min, so the full process of implementing the flight and detection described above came down to ca. 20 min, and two people were sufficient to operate it (i.e., the drone operator and the analyst operating the computer).

An analogical search of a 3-hectare area conducted by a team of rescuers, for example, with the use of the quick-three method (this method assumes that rescuers are divided into groups of three, and one of the rescuers in each group is equipped with a device with a GPS receiver), would take much longer, and the task would have to be carried out by a significantly higher number of people, which entails an increase in the cost of the action.

The system estimates the field coordinates of indicated people with a high accuracy and allows the generation of clear map reports that can be displayed on any stationary or mobile device. After the detection, the analyst transmits the map report to the rescuers, giving them a chance to quickly reach the location of the missing person.

Full field operability means that the detection process can take place entirely in the field, the calculations are performed on a mid-range laptop computer that does not need internet access and communication with high computing power servers.

What is more, the SARUAV solution is not related to any specific UAV platform, so its use in the work of rescuers does not represent a substantial financial barrier. On the contrary, using a SARUAV system can become an element that allows savings if it is a tool that complements standard search methods [23].

However, the newly introduced software does not have the capability of analysing infrared images and videos or to work at night.

The two SARUAV detection algorithms operate in parallel, only on nadir and RGB images. Tests conducted in the near infrared (NIR) gave less satisfactory detection results. It also seems to be inefficient to work on images taken in the far infrared–thermal imaging, as the resolution is insufficient from the viewpoint of algorithms.

SARUAV detection algorithms work on high-resolution RGB imagery, and consequently, their application for night-time images is somewhat limited. Even if night flights were possible from a legal perspective, the images acquired during such a flight are often of insufficient quality for detection to be possible.

Exceptions may be made for areas that are very well lit by artificial light, or for flights with an additional powerful light source mounted on the aerial unmanned vehicle. Such a reflector could be directed downward to illuminate the terrain being photographed, and flights should take place at an altitude that allows appropriate illumination while

contemporaneously maintaining all safety conditions. SARUAV already carried out the first positive tests of this solution.

The cost of the equipment used is as follows:

SARUAV is an innovative IT system to support the search for missing persons—an annual licence for one province costs EUR 1869, including VAT.

The AUTEL EVO II UAV drone is a cost ranging from EUR 1500 to 7000, including VAT.

DJI Mavic II Enterprise Dual UAV drone costs around EUR 3000, including VAT.

To be able to point out a practical application of the above-described software, ten full search flights were analysed, which involved between 14 and 145 nadir aerial images. A review was then carried out of a total of 668 images taken with the use of UAV-mounted cameras. These included missions carried out in both lowland and upland sites. The areas monitored varied in coverage: from simple detections in grasslands and wastelands, to pinpointing people in areas with varying coverage over a small area, to difficult detections in mixed forests outside the vegetative period.

Table 2 provides a tabular summary of the results (the human detector was not familiar with the 668 images previously entered).

**Table 2.** Effectiveness of the SARUAV system depending on relief and land cover.

| Flight No. | Land Relief | Land Cover [1] | Number of Photos | Number of Persons | | Effectiveness [%] |
|---|---|---|---|---|---|---|
| | | | | **Actual** | **Detected** | |
| 1 | upland | a | 37 | 3 | 3 | 100 |
| 2 | upland | a | 124 | 1 | 1 | 100 |
| 3 | upland | a | 98 | 1 | 1 | 100 |
| 4 | upland | b | 115 | 8 | 7 | 87.5 |
| 5 | lowland | c | 20 | 7 | 7 | 100 |
| 6 | lowland | c | 20 | 7 | 6 | 85.7 |
| 7 | upland | d | 14 | 3 | 3 | 100 |
| 8 | upland | d | 18 | 3 | 3 | 100 |
| 9 | lowland | d | 77 | 6 | 6 | 100 |
| 10 | upland | e | 145 | 31 | 30 | 96.8 |
| | | Σ | 668 | 70 | 67 | 100 |

[1] a—temperate broadleaf and mixed forest (lack of leaves outside the vegetative season), b—temperate broadleaf and mixed forest, meadow, developed areas, football field, c—fallow, low vegetation, single trees without leaves, d—meadow, e—temperate broadleaf and mixed forest (lack of leaves outside the vegetative season), meadow, castle. Source: SARUAV [23].

It is recommended that in operational activities, photogrammetric flights with high lateral and longitudinal coverage (at least 60/80%) be executed with the use of the SARUAV system to serve as a basis of further analyses. This means that the person being searched for can be registered in multiple images. The reason why this solution is proposed is because it reduces the probability that a person could remain undetected if he/she is not visible from certain camera positions (e.g., at the border of a meadow and a forest).

The column 'effectiveness' contains a summary of the percentage of detected persons in relation to the actual number of persons out in the wilderness, where the detection of a person is considered to be effective if the SARUAV system detects automatically at least one image covering his/her location in the field (or potentially covering if the person is obscured in some images).

Ref. [24] specified cases of application of the developed methodology in operational conditions that resemble real ones. As part of the research work, several scenarios were worked out of searching for missing persons in mountain and lowland environments, taking into consideration sites with different characteristics (exposed, covered with vegetation or snow). At this point, it should be emphasised that the system was subjected to critical verification by its testing in simulated operational conditions similar to actual ones. The identified problems provided invaluable help in developing the final form of the system (for the production phase). The most valuable conclusion that arises from this study is that

the use of the system makes it possible for rescuers to find a missing person in an open area within just 1 hour. The conclusions formulated at the end of this publication provide a very specific summary of the design and practical functionality of the SARUAV system.

## 4. Discussion and Summary

### 4.1. Drone Module

Technological advancements make it possible to deploy UAVs during various types of search and rescue operations, as well as during natural disasters. As demonstrated by actions executed by the police, firefighters and voluntary organisations, the use of UAVs in search operations has contributed to reducing the time needed to find missing persons [8]. The use of UAVs in search and rescue operations can help achieve a significant reduction in the number of victims of various types of accidents, and through the use of avalanche victim detectors, they can also provide support during search operations after avalanches have descended [25].

Adequate technological equipment of drone modules, as well as good coordination in operational activities and developed cooperation of various services (state and voluntary), will contribute to increasing the effectiveness of search operations. It should also be pointed out that cooperation between different formations should include an increase in the number of specialist training courses to deepen theoretical and practical knowledge (along with an analysis of actions taken), as well as joint search manoeuvres in various areas and terrains. The practical use of knowledge and experience could significantly affect the system of searching for missing persons and increase the effectiveness of operations.

### 4.2. UAV Ground Control Station

Members of the VFD designed and executed a GCS workstation dedicated to actions involving the use of rotary wing UAVs. The tested system allows streamlining the communication between HRO, UAV section and search participants, and thanks to the possibility of radio signal amplification and the transmission of necessary data, it is also appropriate for searching people in mountainous areas. Before now, when this solution was not available yet, problems arose in obtaining radio communication due to the demanding and difficult terrain owing to the presence of rocks and ridges, as well as deposits of various deposits. Since radio communication was supported by relay stations, a significant improvement in signal quality has been achieved.

### 4.3. SURUAV System

The undertaken operational activities could be supported by the use of SARUAV software, the effectiveness of which was proven during a rescue operation conducted in a mountainous terrain with large forest complexes in the Low Beskid (June 2021). While searching for a lost male subject aged 65 years (the person in need of assistance was ailing and was not carrying a phone) first of all, traditional search and rescue methods were used. After the lapse of more than 20 h, the decision was made to provide support by using a specialised drone and the SARUAV system. As a result of this solution, after 4 additional hours, the rescue team was able to find the lost subject.

Figure 6 shows a photo in which the SARUAV software identified the missing person (this would have been very difficult when visually reviewing multiple photos, and the further difficulties were such factors as tall, two-metre-high grass and the passage of a storm, after which the dogs lost the trail).

The software automatically detects people in aerial photos by indicating places where a person could potentially be—it carries out aerial human detection. The algorithm sets out the search site, where the UAV takes hundreds of photos. Thanks to the software, several hundred photos are analysed in only a few minutes, which would be absolutely impossible for the human eye to analyse, particularly given such quantities, maintaining this kind of speed and accuracy—it would simply not be able to notice the missing person. When a person is detected by the SARUAV system, a location pin is placed on the map,

and the UAV operator then knows exactly where to direct the rescuers. Such a solution is innovative on a global scale because this technology allows a significant reduction in time needed to find a missing person during search and rescue operations, which clearly has an advantageous impact on the further health and life of the victim. In its analysis, the SARUAV system can take into account such inconspicuous elements as a hat lying on the ground (which can be an excellent indication in the case of a real search) or an image of a dog running alone (its owner may be present close by). Works are underway to adapt the system for search operations on water.

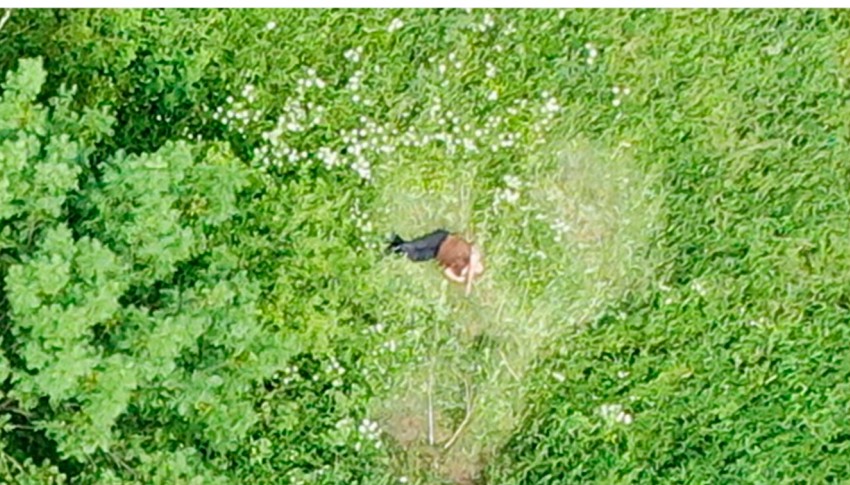

**Figure 6.** Image processed by the SARUAV application indicating the missing person. Source: [15].

Exercises conducted at the MSFS Field Training and Rescue Innovation Base have shown that access to specialised equipment and modern solutions is a guarantee of a shorter time that would be required to achieve the intended objective.

The use of UAVs is proven to facilitate efficient mapping of the search area, followed by coordinated operations using a GPS and compass module, and in an emergency situation by relaying messages and commands from the HRO to teams and sections via a speaker integrated into the drone.

One of the most common technologies in AI, the so-called neural networks, can be used to support this type of activity. The algorithm has several inputs via which information is received, and an "inference module" that generates an output signal on the basis of input information and their weights. The processed information is then directed to the output and passed on. In this way, an image seen by a camera, for example, can be compared with a previously created pattern. These patterns can be further developed, and machines can be taught new patterns through the process of machine learning.

Such a solution that involves the SARUAV system is already available for implementation by the police, fire and border guards, as well as mountain and water rescue units. Its functioning requires basic information, such as the last known location of the lost person.

The availability of the SARUAV module during tests and real search actions would make it possible to extend the activities performed by an automatic analysis of aerial photographs and searching for a missing person on them. The system has high detection efficiency and successfully distinguishes human silhouettes from other elements of the environment, such as animals, vegetation or rocks [26].

## 5. Recommendations

1.  The application of the above outlines the organisational scheme of the UAV sections along with dedicated solutions to support equipment communication, and team liaison allows UAVs to be launched within 3 min of arrival in the operational area. The first image in the GCS workstation can be obtained already in the 6th minute, and

the image in the mobile devices of the participants of the operation in the 8th minute after the commencement of the operation.

2. The use of SARUAV software allows replacing human labour associated with time-consuming analyses of images taken from the air with automatic image recognition. In combination with the possibility of covering the area of operations by a larger number of UAVs, this provides an extremely efficient system for the search for missing persons.

3. Search and rescue activities should be oriented at assuring that the ability to carry out rescue operations on a basic level could become universal for all NFRS entities [27].

4. The nationwide cooperation of the police with search and rescue groups and other entities should be further intensified and developed, involving, inter alia, the launching of joint undertakings (training, sham search operations) the purpose of which comprises exchanging experience, mutual requirements and consolidation of knowledge in the search for missing persons [28].

5. In the area of scientific development in the context of search and rescue operations, it is necessary to conduct research to reduce the time it takes to find a missing person. In this respect, there is a need to improve the technical parameters of components of search coordination systems:

   - UAVs—increase in flight duration;
   - Data transmission devices—increased speed of information transfer (including high quality images);
   - GCS workstation—extending capabilities by subsequent modules (currently: ensuring stable communication in the official SFS radio bands, creation of own Wi-Fi network in the MESH system, preview on the monitor and transmission of the image from UAVs to the participants of the action in possession of any mobile device, acting as a web server with important data concerning the conducted action, and strengthening of mobile telephony signal).

6. The conducted research related to the organisation of search and rescue operations confirmed the following:

   (a) The need to develop cooperation between rescue parties and improve coordination of actions undertaken on a previously unknown terrain;

   (b) The benefits of setting up a command post in the vicinity of the incident and starting to support the HRO by viewing the situation from above;

   (c) In the case of a lack of communication, the solution to the problem turned out to be GCS workstation being on the equipment of the UAVs VFD section, which uses antennas that are much more robust.

## 6. Conclusions

Unmanned aerial vehicles (UAVs), commonly known as drones, are becoming increasingly common and keep gaining new functionalities. Currently, they are one of the most innovative elements in the activities of rescue services, including the State Fire Service (SFS).

On 8 July 2021, on the premises of the MSFS Field Training and Rescue Innovation Base in Nowy Dwór Mazowiecki, tests were conducted to verify the characteristics, specifics and prospects of using drones in rescue operations. During the exercises, the hardware and software used by VFD were tested as well as the possibilities of using UAVs and GCS workstation.

The scheduled exercise scenarios included terrain mapping—producing a map under field conditions, coordination of ground groups, evaluation of the terrain (object fire) with the use of a thermal imaging camera, and precise observation of a designated object (region and person).

The research participants were involved in various simulated actions. They coordinated the action, using mounted speakers and the UAVs preview. The aim of joint exercises was to achieve an improvement of techniques and skills of coordination of actions and cooperation.

Moreover, the article also presents aspects related to the use of unmanned aerial systems supported by image recognition software based on artificial intelligence algorithms. The SARUAV system of searching for missing persons, dedicated in particular to open spaces, the main part of which is the algorithm of detecting silhouettes of people from images acquired from UAVs, is being used in Poland by 4 VFD units and 2 MVRS groups. The SARUAV solution was developed and tested in cooperation with the MVRS Jurassic Group. Field tests of the system have pointed to a high performance of the algorithm, and the results were published in [24,29,30]. The system received very positive feedback from, among others, MVRS, but also from the FlyTech UAV company from Krakow, which produces professional BIRDIE UAVs that can be used in the SAR rescue system.

The system was also proven effective in a real-life situation, where time was of the essence.

**Author Contributions:** Conceptualisation, N.T. and W.W.; methodology, W.W.; validation, W.W. and N.T.; formal analysis, N.T.; investigation, N.T.; resources, W.W.; writing—original draft preparation, N.T.; writing—review and editing, W.W.; visualisation, N.T.; supervision, W.W. All authors have read and agreed to the published version of the manuscript.

**Funding:** This research received no external funding.

**Institutional Review Board Statement:** Not applicable.

**Informed Consent Statement:** Not applicable.

**Conflicts of Interest:** The authors declare no conflict of interest.

## Appendix A

**Table A1.** List of equipment used in the exercises [1].

| Type of Equipment | Additional Equipping | Operating Limitations |
|---|---|---|
| UAV AUTEL EVO II | 4 battery packs Live Deck transmission systemControl unit Battery charging station | Weather without precipitation Temperature from −10 to +40 °C Flight duration 25–38 min on one battery (depending on weather) Wind speed < 40 km/h Battery charging time approx. 80 min |
| UAV DJI Mavic II Enterprise Dual | 3 battery packs Spotlight Loudspeaker Thermal imaging camera Battery charging station | Weather without precipitation Temperature from −10 to +40 °C Flight duration 18–25 min on a single battery (depending on weather conditions) 18 min—using the spotlight 20 min—using the loudspeaker Wind speed < 40 km/h Battery charging time approx. 80 min |
| Ground Control Station Workstation | Motorola transceiver 4600e (mobile) + antenna with high energy gain Wi-Fi network LTE network 19″ monitor for viewing live from UAV AUTEL Redistribution of images to mobile devices Power distributor for 4 socketsMast (tripod) for installation of antennas—at a distance of up to 10 m from GCS workstation | A 230 V alternating current source is required Power (depending on the number of receivers) approx. 1000–1500 W Weather without precipitation or use in a sheltered location Temperature from −18 to +45 °C |

**Table A1.** *Cont.*

| Type of Equipment | Additional Equipping | Operating Limitations |
| --- | --- | --- |
| ACER Laptop | SD card reader<br>Wireless mouse<br>230 V power supply (included in estimated power of GCS workstation)<br>Ability to view live image from UAV | Weather without precipitation or use in a sheltered location<br>Temperature from −10 to +40 °C |
| Samsung Tablet A10—for the operation UAV | USB-C power adapter | Weather without precipitation |
| 5 pcs. Motorola GP360 Radiotelephone for communication within the group of UAV operators | Chargers | Temperature −18 to +45 °C |

[1] Source: [20].

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
