# Peer review of "The Efficiency of Drones Usage for Safety and Rescue Operations in an Open Area: A Case from Poland"

_sustainability, doi:10.3390/su14010327_

Round 1
Reviewer 1 Report
The general idea of this article is interesting and can be very useful. The manuscript is well is well-written and easy to follow.
The authors are presenting the outcomes of the SARUAV software in the results section. It is not clear if the authors used this software in their experiment or these are results of the sofware applied in another case study. The authors should clarify whether these results are from their research or not.
Additionally, it is not explained/presented how the SARUAV detection algorithmic process work. In what algorithmic model is based. How is the system trained? What is capable of detecting? In what data is the detection working? Can thermal data be used as input on the software? Furthermore, the authors should explain how they used it in their experiment in Nowy Dwór Mazowiecki. These are fundamental aspects of the UAV SAR application and must be added to conclude on the added value of the methodology compared to the tools already available.
Also, abbreviation use, spacing and typing errors, figure clarity should be improved. As for figures, the map in the left part of Figure 1 has small letters, and thus, the names are not readable.
Author Response
In the experiment carried out in Nowy Dwór Mazowiecki, during testing phase selected systems supporting search and rescue operations (UAVs, command suitcase) were used. For formal reasons, the SARUAV system was not used for testing (which was tried to be shown in the study), and a concept was presented that extended the functionality of the coordinated search and rescue group with the SARUAV system. We found some license restrictions in one province due to the lack of availability of appropriate digital maps. For each launch of the system, it is necessary to prepare dedicated spatial data, and the automatic detection of people works in a specific area "purchased" by the recipient of the system. Evidence of the effectiveness of the presented concept with the SARUAV model has been demonstrated in the form of positive results of finding a living person and may constitute an effective extension of the functionality of the system used in search and rescue operations: in [Niedzielski, T .; Jurecka, M .; Mizinski, B .; Pawul, W .; Motyl, T. First Successful Rescue of a Lost Person Using the Human Detection System: A Case Study from Beskid Niski (SE Poland). Remote Sens. 2021, 13, 4903. https://doi.org/10.3390/rs13234903].
The solution uses a nested k-means algorithm to detect people in aerial photos from close proximity [Niedzielski, T .; Jurecka, M .; Stec, M .; Wieczorek, M .; Miziński, B. The nested k-means method: A new approach for detecting lost persons in aerial images acquired by unmanned aerial vehicles. J. Field Robot. 2017, 34, 1395–1406].
The software uses a variety of non-statistical detection methods [Niedzielski, T.; Jurecka, M.; Miziński, B.; Pawul, W.; Motyl, T. First Successful Rescue of a Lost Person Using the Human Detection System: A Case Study from Beskid Niski (SE Poland). Remote Sens. 2021, 13, 4903. https://doi.org/10.3390/rs13234903].
The SARUAV system was trained on pictures of people wearing clothes of different colors and patterns (including smaller objects: children and dogs on the pictures). The training sets were made with engagement of the study participants.
The SARUAV system effectively detects the figures of any dressed adults and children, and the presence of dogs (it does not detect big animals).
The key parameter for the SARUAV system is the ground sample distance (GSD), which depends on the flight altitude and the characteristics of the camera. It is best to fly so that the GSD is less than 3 cm / px.
The two detection algorithms work in parallel, only on nadir and RGB images. The conducted near infrared (NIR) tests gave less satisfactory detection results. From the perspective of the algorithms used, the far-infrared imaging has insufficient resolution.
The SARUAV system was not used in the experiment in Nowy Dwór Mazowiecki. A new research is planned with a system in a different geographic area. This system is a conceptual proposal of the authors of the article and its effectiveness will be presented in the next study.
The drawing has been replaced with an image from google maps (satellite view of the operation area).
Reviewer 2 Report
The respond is OK.
Author Response
Thank you for the support and comments that have enriched the content of the publication.
Reviewer 3 Report
Dear author, First, thank you for your response to my comments. It seems that you performed so many efforts with the correction of your study. Thank you for your efforts with revising the manuscript. Congratulations.
The previous suggestions and comments on the first version have been seemed to be considered in detail. Considering the first version of your manuscript, I believe that this performed revision helped to improve your paper. As mentioned earlier, your title has an interesting theme to save humanity. So, I wish you success in your future projects. I only have a small suggestion to pay attention to writing style. For example, look at line 590 "(region, person)". It can be write as "region and person". Regards
Author Response
We would like to thank you for all comments, which improved the quality of the publication. Final editing corrections have been implemented.
Round 2
Reviewer 1 Report
The authors fulfilled all my previous comments in this version of the manuscript. They added a paragraph (lines 112-134) presenting how the SARUAV system works and what it can detect. They clarified that it is not used in their experiment due to license restrictions and lack of "availability of appropriate maps." Additionally added information on how the SARUAV detection process works and in. Thus, the manuscript can be published in the Sustainability journal.
This manuscript is a resubmission of an earlier submission. The following is a list of the peer review reports and author responses from that submission.
Round 1
Reviewer 1 Report
The general idea of this article is interesting and can be very useful. However, the authors need to clarify fundamental aspects of the methodology used in order to be able to conclude on the added value that the methodology presented has compared to the methods already available.
The relationship of this paper to other similar research is not sufficiently discussed. Also, term consistency (for example UAV / UAS), abbreviation use (VFD, HRO, SARUAV ect), spacing and typing errors, figure clarity and caption should be improved.
To the reviewer, the paper has no scientific contribution, and it is merely a technical report of the followed steps and the configuration used on the three scenarios. Moreover, the claimed contributions in using UAS in SAR operations are very well-known by the drone specialists especially those using UAS in SAR.
Additionally, the bibliography of the manuscript should be improved as 9/19 of the references are reports or web resourced from press and/or websites.
Following are few relevant references
- Weldon, W.T.; Hupy, J. Investigating Methods for Integrating Unmanned Aerial Systems in Search and Rescue Operations.Drones 2020, 4, 38. https://doi.org/10.3390/drones4030038
- Półka, Marzena & Ptak, Szymon & Kuziora, Łukasz. (2017). The Use of UAV's for Search and Rescue Operations. Procedia Engineering. 192. 748-752. 10.1016/j.proeng.2017.06.129.
- Silvagni, A. Tonoli, E. Zenerino, M. Chiaberge, Multipurpose UAV for search and rescue operations in mountain avalanche events. Geomatics Nat. Hazards Risk 8, 18–33 (2017).
Also, the paper has many questionable claims or statements:
-Why they authors are presenting SARUAV software test results in section 3.5 (lines 267 304) as the software was not possible to be tested (lines 257-258). To the reviewer's understanding, the test results presented are the results of another research as the authors state in lines 268-275.
-Why the authors state in lines (391-392) that “The article presents aspects regarding the use of unmanned aerial systems supported by image recognition software based on artificial intelligence algorithms” this is not clear in the manuscript.
-Why in the conclusion section the authors are presenting the SARUAV solution limitations and results despite the fact they didn’t used it to their research (lines 257-258).
Consequently, I will recommend rejecting the paper.
Reviewer 2 Report
In this paper, the purpose of this research was to analyze the feasibility of using unmanned aerial systems in the search for missing persons in open areas. Tests were conducted on the use of UAVs in search and rescue operations according to three developed scenarios, one of which included simulation of radio communication failure. The results of the tests confirmed that the time of reaching the injured, which is crucial for the prognosis of their survival, can be significantly reduced by using the imagery recorded by UAVs. The effectiveness of the search system was tested by comparing the arrival times of the teams with and without GPS and compass equipment. However, the following problem need to be improved.
(1) The English of paper is poor, please polish it again.
(2) Please highlight the novelty of the proposed algorithm in the introduction.
(3) Please regroup the format of the paper.
(4) More experimental needed to show the advantages of the proposed method.
Reviewer 3 Report
Dear Authors,
Thank you for your effort with this study. Drone-based studies are famous thanks to their innovative technology and low-cost. Safety and rescue is also very important for injured and lost people in wild and mountainous areas. Please find my recommends and suggestions as given below:
Title:
Title is appropriate for your study.
But, it can be improved as The Usage efficiency of drones for safety and rescue operations in the open area: A case from Poland.
Abstract:
Line 16: What is the equivalent of “GCS”? If it is a ground control station or ground control system? Readers can confuse. Can it be a ground control station supported by the workstation?
Please check your abstract. For example, before you get results, you might mention your approach. Look at line 17-18. There is a method sentence that should be moved to line 13 before the result.
Keywords:
It is appropriate. Line 21-22. But abbreviation of unmanned aerial vehicle is enough. You may remove UAV. Please add automatic flight instead of automatic person detection.
Introduction:
Please look at lines 53-55. There should be several opportunities for object detections. Third party software in the markets, which can integrate to UAVs, are available in the literature.
For example, camera-mounted UAVs, which are mostly use in civil or military purposes, have an ability to detect animals or human. So, recent published papers can be found about it.
As a reader, I believe that the quality of introduction will be improved than now. At the discussion section, expanding your introduction part will also give you an opportunity to discuss it with your results and method.
Line 62. Please use at least two sentences for the study aim.
Method:
Please add some information. For example, how many persons controlled the flight operation? What kind of time measurement technique was used? What about the weather conditions? (Cloud ratio, wind speed, temperature…). Brief information about the software architecture?
Results:
Please hesitate to write paragraphs with a sentence. Look at lines 111-112.
Subheadings (3.1, 3.2, 3.3) of results are similar to method! You may move it to method section.
Results:
Your study is a time saver. But please explain the disabilities of your study. For example, can we use your newly introduced software for IR bands instead of RGB? The cost of your employed equipment?
Other parts:
After improve your indroduction, method parts, you need to redesign your discussion and conclusion parts.
General comment:
After reading your paper, I assumed that you developed an innovative method for real-time object detection system, which integrated with a drone.
All abbreviations should be listed or mentioned before. Please check the whole paper for abbreviations.